# Red Blood Cell Fatty Acid Profiles Are Significantly Altered in South Australian Mild Cognitive Impairment and Alzheimer’s Disease Cases Compared to Matched Controls

**DOI:** 10.3390/ijms241814164

**Published:** 2023-09-15

**Authors:** Varinderpal S. Dhillon, Philip Thomas, Sau L. Lee, Permal Deo, Michael Fenech

**Affiliations:** 1Health and Biomedical Innovation, UniSA Clinical and Health Sciences, University of South Australia, Adelaide 5000, Australia; permal.deo@unisa.edu.au (P.D.); michael.fenech@unisa.edu.au (M.F.); 2CSIRO Health and Biosecurity, Adelaide 5000, Australia; phil.t63@hotmail.com; 3College of Medical and Public Health, Flinders University, Bedford Park 5042, Australia; sau.lee@flinders.edu.au; 4Genome Health Foundation, Adelaide 5048, Australia

**Keywords:** red blood cell, fatty acids, Alzheimer’s disease, mild cognitive impairment

## Abstract

Nutritional imbalances have been associated with a higher risk for cognitive impairment. This study determined the red blood cell (RBC) fatty acid profile of newly diagnosed mild cognitive impairment (MCI) and Alzheimer’s disease (AD) patients compared to age and gender-matched controls. There was a significant increase in palmitic acid (*p* < 0.00001) for both MCI and AD groups. Saturated fatty acids were significantly elevated in the MCI group, including stearic acid (*p* = 0.0001), arachidic acid (*p* = 0.003), behenic acid (*p* = 0.0002), tricosanoic acid (*p* = 0.007) and lignoceric acid (*p* = 0.001). n-6 polyunsaturated fatty acids (PUFAs) were significantly reduced in MCI, including linoleic acid (*p* = 0.001), γ-linolenic acid (*p* = 0.03), eicosatrienoic acid (*p* = 0.009) and arachidonic acid (*p* < 0.00004). The n-3 PUFAs, α-linolenic acid and docosahexaenoic acid, were both significantly reduced in MCI and AD (*p* = 0.0005 and *p* = 0.00003). A positive correlation was evident between the Mini-Mental State Examination score and nervonic acid in MCI (r = 0.54, *p* = 0.01) and a negative correlation with γ-linolenic acid in AD (r = −0.43, *p* = 0.05). Differences in fatty acid profiles may prove useful as potential biomarkers reflecting increased risk for dementia.

## 1. Introduction

It has long been regarded as a normal part of ageing that an individual’s memory capacity may decline with advancing years [1,2,3]. Over the past few decades, several studies have shown that a subset of individuals experience a level of memory loss greater than normal and that the incidence of this condition is increasing [4,5]. MCI has a prevalence of up to 10–20% of the population aged over 65 years and is clinically defined as cognitive impairment beyond that expected for an individual’s age and education [2,5]. It has been shown that MCI individuals, when followed longitudinally, have an increased risk of conversion of 10–15% per year to AD compared to normally ageing individuals who have a conversion rate of 1–2% per year [6,7,8]. AD is characterised clinically by a gradual and progressive degenerative loss of memory, visuo-spatial and speech disorder and various adverse mental health changes [9,10,11,12,13]. The global incidence of individuals living with AD is currently greater than 36 million, with a predicted increase towards 115 million by the year 2050 [14]. In Australia, it is estimated that 401,300 individuals are living with AD, with $3 billion spent on health and aged care directly for dementia in 2018–2019, hence imposing a huge social and economic burden [15].

Nutritional deficiencies or excesses of both micronutrients and macronutrients have been associated with cognitive decline and have been proposed to be potential risk modifiers for cognitive impairment [16,17,18,19,20]. Fatty acids are necessary for brain development and comprise up to two-thirds of the dry weight of the human brain, with one-third of the lipid content consisting of PUFA, such as arachidonic acid (C20:4n-6) and DHA (C22:6n-3) [21,22,23]. A decrease in DHA plasma levels has previously been shown to be a risk factor for developing AD [24,25,26]. Fatty acid profiles within the brain and other tissues may have a significant impact on the biophysical properties of cellular membranes via modulation of ion channels as well as impacting membrane fluidity [25]. It has been shown that DHA exerts a key role in influencing the fluidity of cellular membranes as well as long-term potentiation of neuronal signal transmission, one of the major cellular mechanisms underlying the ability to remember [27,28]. Earlier studies have shown that a higher intake of monounsaturated fatty acids (MUFAs) via olive oil in the Mediterranean diet [29,30], along with a lower intake of foods that are rich in saturated fatty acids (SATFA) [31,32], may contribute to decreased dementia risk [33,34]. However, verification of these observations requires direct evidence by investigating the lipid content of various tissues in vivo.

The aim of the present study was to find out the RBC fatty acid profile within individuals newly diagnosed with MCI or AD and compare their profiles in relation to each other and age and gender-matched controls, based on the hypothesis that these profiles would be significantly different. The profiles consisted of SATFA, monounsaturated fatty acids (MUFA), PUFA, as well as ratios of n-3:n-6 PUFA. Red blood cells were the selected tissue because their fatty acid composition reflected the dietary fatty acid intake over at least 4 months (average life span of the RBC), whereas plasma concentration is less reliable because it can vary within and between days depending on daily variation in food intake [35,36]. Harris and co-workers have shown that the composition and concentration of fatty acids within the RBC membrane, especially that of omega-3 fatty acids, strongly correlated with that in cardiac tissues and are hence more biologically relevant as a surrogate tissue to infer lipid composition in various internal body organs [37]. Furthermore, RBC levels of fatty acids are associated with brain structure and function [38,39,40].

## 2. Results

The analysis showing the percentage of the total RBC fatty acids is presented in Table 1. For SATFA, a one-way ANOVA analysis showed a significantly higher proportion of palmitic acid (C16:0; *p* < 0.00001) was found in both MCI and AD individuals compared to the control group. A Significantly higher proportion was found for stearic acid (C18:0; *p* = 0.0001), arachidic acid (C20:0; *p* = 0.003), behenic acid (C22:0; *p* = 0.0002), tricosanoic acid (C23:0; *p* = 0.007) and lignoceric acid (C24:0; *p* = 0.001) for the MCI group only. These results are outlined in Table 1.

For MUFA, oleic acid (C18:1n-9) was the only fatty acid found to be significantly elevated in both the MCI and AD groups (*p* = 0.002; Table 1).

For PUFAs, a significant reduction was found for α-linolenic acid (C18:3n-3; *p* = 0.00106) and DHA (C22:6n-3; *p* = 0.00003) in both MCI and AD groups. Significant reductions were also evident for linoleic acid (18:2n-6; *p* = 0.001), γ-linolenic acid (C18:3n-6; *p* = 0.03), eicosatrienoic acid (C20:3n-6; *p* = 0.009) and arachidonic acid (C22:6n-3; *p* = 0.00004) for the MCI group only. The total PUFA content was significantly lower only within the MCI group compared to the control group (*p* < 0.00002). There was also a significant reduction in total n-3 fatty acids (*p* = 0.0002) and total n-6 fatty acids (*p* = 0.00003) in the MCI group. The n-3/n-6 ratio was significantly reduced for the MCI group but not the AD group when compared to the control group. These results are outlined in Table 1.

When the fatty acid profiles were directly compared between the MCI and AD study groups, there was a significant increase in five of the SATFAs, including stearic acid (C18:0), arachidic acid (C20:0), behenic acid (C22:0), tricosanoic acid (C23:0) and lignoceric acid (C24:0) in the MCI group relative to the AD group. There was also a significant decrease in arachidonic acid (C20:4n-6), docosapentaenoic acid (C22:5n-3), total n-6 PUFA, total PUFA and PUFA/SATFA (ratio) as well as an increase in total SATFAs in the MCI group compared to the AD group. These results are outlined in Table 1.

Pairwise comparisons for MCI and age and gender-matched controls and AD and age and gender-matched controls are outlined in Table 2. In addition, pairwise comparisons for both individual and combined gender for MCI and age and gender-matched controls and AD and age and gender-matched controls are outlined in Appendix A. Compared to the matched controls, the MCI group had higher values for the SATFAs palmitic acid (*p* = 0.001), stearic acid (*p* = 0.04), tricosanoic acid (*p* = 0.02), lignoceric acid (*p* = 0.03) and total SATFAs (*p* = 0.004), a difference that was observed only in females. For the AD group, arachidic acid was significantly increased when compared to age-matched controls, but this was only observed in females (*p* = 0.003). For MUFA, there were no significant differences between males and females within the MCI group when compared to matched controls. Compared to matched controls, the AD group had higher oleic acid and total MUFA, but this finding was observed only in females (*p* = 0.002 and *p* = 0.02, respectively). For PUFA, there was a significant reduction in γ-linolenic acid (*p* = 0.02), α-linolenic acid (*p* = 0.005), eicosatrienoic acid (*p* = 0.04), arachidonic acid (*p* = 0.004), eicosapentaenoic acid (*p* = 0.02) and DHA (*p* = 0.006), as well as total PUFAs (*p* = 0.004) within the MCI group when compared to controls, but the findings were observed only in females. A number of PUFAs were significantly reduced in the AD group, including α-linolenic acid (*p* = 0.04), arachidonic acid (*p* = 0.01), DHA (*p* = 0.004) and total n-3 PUFAs (*p* = 0.04) when compared to matched controls, but the findings were evident only in females. The above results are outlined in Table 2 and Appendix A.

ROC Curves were generated (along with sensitivity, specificity and likelihood ratios) for RBC fatty acids that were found to be significantly different between the groups, and results are represented in Appendix A. For the MCI group, the AUC was greater or equal to 0.80 for 3 of the 6 SATFAs measured (behenic acid (C22:0; AUC = 0.80), stearic acid (C18:0; AUC = 0.80) and palmitic acid (C16:0; AUC = 0.85), two n-6 PUFAs, γ-linolenic acid (C18:3n-6; AUC = 0.81) and arachidonic acid (20:4n-6; AUC = 0.84) and two n-3 PUFAs, α-linolenic acid (C18:3n-3; AUC= 0.81) and DHA (C22:6n-3; AUC = 0.83)).

For the AD group, palmitic acid (C16:0) had an AUC of 0.75, oleic acid (C18:1n-9) had an AUC of 0.79, linolenic acid had an AUC of 0.71, and for DHA the AUC was 0.67. Graphs showing AUC of 0.8 and above and *p*-values for palmitic acid, stearic acid, γ-linolenic acid, behenic acid, arachidonic acid, linolenic acid and DHA acid in the MCI group are shown in Figure 1. Overall, 13 fatty acids were significantly different relative to healthy controls in MCI, but in AD, only 4 were significantly altered, and their ROC AUC was generally less compared to MCI.

Cross-correlation analysis between the Mini-Mental State Examination (MMSE) score and individual components of the RBC fatty acid analysis showed a positive correlation for the MMSE score with nervonic acid (C24:1n-9) in MCI individuals (r = 0.54, *p* = 0.01) and a negative correlation for the MMSE score with γ- linolenic acid (C18:3n-6) in AD individuals (r = −0.43, *p* = 0.05).

## 3. Discussion

MCI is a condition by which an individual has a greater rate of cognitive decline for an individual’s age and education level but does not negatively impact the activities of daily life. This is in stark contrast to individuals with AD, whose cognitive deficits are more severe and widespread and substantially affect daily function. MCI can therefore be seen as a risk state for dementia [5,41]. The aim of the study was to investigate the RBC fatty acid profile of newly diagnosed individuals with MCI and AD from South Australia and compare them to each other and to healthy individuals. The results of the present study support the hypothesis that the lipid profiles are significantly different between these three study groups. Previous epidemiological studies have suggested that elevated levels of SATFAs may be associated with long-term cognitive decline [22,42,43,44,45]. However, a recent study from Italy showed that participants who consumed a moderate amount of short-chain saturated fatty acids (SCSFAs) and medium-chain saturated fatty acids (MCSFAs) have a lower risk of cognitive impairment [46]. In the current study, we found 6 of the 8 SATFAs investigated to be significantly elevated within the MCI group and, in the case of palmitic acid, to be significantly increased within the AD group. These findings reflect the results of previous studies that have found that levels of high dietary fat intake, SATFAs, in particular, are associated with a deterioration in cognitive function [32,47,48,49,50,51,52]. A study from the Netherlands examined the fatty acid profile in RBCs from mild AD cases and healthy controls [53] have also reported significant increases in various fatty acids, such as palmitic acid, oleic acid, and significantly lower concentrations of γ-linolenic acid, α-linolenic acid, arachidonic acid and DHA, and is in agreement with our findings in MCI and AD. These observations indicate the possible robustness of using RBC lipid profiles as a potential biomarker for dementia risk.

The precise biological mechanisms underlying the relationship between elevated SATFAs and their effects on an individual’s cognitive impairment have not been clearly elucidated. It has been proposed that SATFAs may not directly influence brain function but may affect it indirectly through modulating risk factors for various chronic diseases, such as cardiovascular disease or type 2 diabetes, thereby influencing cognitive decline [54,55]. Greenwood et al. have shown that in both animal and human studies, high SATFA diets are associated with a higher risk of developing dementia in individuals who express insulin resistance or have type 2 diabetes mellitus [55]. It has also been shown in the transgenic mouse model (APPswe/PS1dE9) for AD that following a SATFA diet for 3–4 months increases Aβ levels in the hippocampal region compared to a diet containing PUFAs [56]. A previous study involving rats showed that those animals who were given a high-fat diet for 4 months led to significantly higher levels of free radical production, NADPH oxidase and significantly increased Cox-2 expression in the cerebral cortex region of the brain. These findings suggest a generation of an inflammatory response involving NF-K (a hallmark of oxidative stress) [57]. Oxidative stress and inflammation are significantly prevalent in AD aetiology, thus offering a further link between intake of diets high in dietary fatty acids and increased risk for developing dementia [43,57,58]. Furthermore, rodent studies which involved being placed on a high-fat diet for 2 months showed significantly low levels of hippocampal brain-derived neurotrophic factor (BDNF), resulting in lower neuronal plasticity and spatial learning output [59].

In this study, the total MUFA levels were not significantly different between the three study groups. However, of the four MUFAs analysed, only oleic acid (C18:1) concentration was significantly higher in the MCI and AD group compared to the healthy individuals. Our results are conversant to the findings from a 3-year follow-up study that showed MUFAs as having neuroprotective properties resulting in reduced cognitive impairment [60]. However, the findings from the latter study were based only on food frequency questionnaire (FFQ) data on the dietary intake of fatty acids, which may explain the different outcomes from our study based on RBC fatty acid measurement. A recent study involving a middle-aged group population showed that the total MMSE score was negatively associated with age and plasma SFA levels but was positively associated with MUFAs [45]. Furthermore, oleic acid, which was the MUFAs that increased in both MCI and AD in our study, is the cis-isomer of elaidic acid, which is the most abundant trans-fatty acid in the human diet (especially in junk food), which has been shown to generate dysfunctional high-density lipoprotein (HDL) and induce hyperlipidemia and non-alcoholic steatohepatitis by upregulating the expression of proteins required for cholesterol synthesis and its transport [61,62,63,64]. Although most naturally occurring fatty acids are in cis-form, fatty acids may be converted to trans-forms by hydrogenation [65]. Therefore, it is possible that the association of oleic acid with MCI or AD may be confounded by the presence of elaidic acid, which is found in margarine, partially hydrogenated oils and fried foods [66]. A recent study in Japan showed that serum elaidic acid concentration predicted a high risk of all-cause dementia [67]. It has also been suggested that the health effect attributed to MUFAs may be reflective of other dietary factors that are more efficacious in protecting against cognitive impairment, such as olive oil, which also contains higher levels of antioxidants and numerous polyphenols, such as oleuropein [43,68,69]. Within this Australian study, it would appear that the levels of MUFAs are possibly insufficient to derive any beneficial effect relating to brain health at the vascular or neuronal level. Therefore, future research in this area should address these questions: (i) whether oleic acid or elaidic acid is harmful to brain function, and (ii) to investigate the mechanisms involved in MUFA intake and cognitive decline. It would be interesting to test whether oleic acid alters the LDL/HDL ratio in blood or affects neuronal transmission or synaptogenesis in MCI and AD cases.

This study also analysed the RBC fatty acid PUFA profile and found that the levels of 6 out of 7 PUFAs analysed in MCI and 2 out of 7 in AD were significantly reduced and also resulted in a reduced n-3/n-6 PUFA ratio. These results reflect findings from previous studies that have also found significant changes within the PUFA profiles of individuals with cognitive impairment [70,71,72,73]. It has been shown previously within a number of epidemiological studies that maintaining sufficient levels of PUFAs and, in particular, DHA and precursors of n-3 synthesis, such as α-linolenic acid, may be protective against age-related cognitive decline or dementia, including AD [32,52,70,74,75,76,77,78,79,80]. The potential protective effects of PUFAs in relation to cognitive impairment may be related to lowering the risk for heart diseases [70,81] or non-haemorrhagic stroke [82], which has been shown to escalate the risk of developing dementia and its major subtypes [83,84]. It has also been shown that by alleviating the synthesis of pro-inflammatory cytokines, PUFAs may impair the pro-inflammatory pathways that are associated with the aetiology of cognitive impairment and, therefore, reduce the advancement of the disease [85,86,87]. PUFAs, such as DHA, are primary components of cerebral membrane phospholipids; a reduction in DHA could, therefore, compromise both membrane integrity and neuronal function maintenance, thus alleviating any potential protective effects [83]. It has also been suggested that omega-3 fatty acids may alter membrane stability as well as amyloid precursor protein (APP) cleavage by β-secretase, thereby reducing the production of beta-amyloid (β-amyloid) [88,89,90]. It has also been shown that rats fed on a high PUFA diet showed a 10-fold increase in transthyretin expression, which binds to β-amyloid and, therefore, may enhance the clearance of β-amyloid from the brain [91]. Transgenic mice exhibiting amyloid and tau pathologies when fed a diet supplemented with DHA for 9 months have shown significantly reduced intra-neuronal accumulation of both β-amyloid and tau proteins as a result of decreased presenilin-1 levels rather than altered enzymatic APP cleavage by α- or β-secretases [92]. 

Our study also showed definite gender effects in relation to fatty acid profiles. In the female MCI sub-group, significant increases were found in three SATFAs relative to the MCI control group, in line with a recently published study [93]. Conversely, four of the PUFAs measured were significantly lower in MCI females compared to the MCI control group. Similarly, significant results were consistently found in the female MCI sub-group with a significantly higher level of total SATFAs while significant decreases in total omega-3 FAs, total omega-6 FAs, the n-3/n-6 FA ratio, omega-3 index and total PUFA/total SATFA ratio were evident relative to the MCI control group. For the AD group, when compared to age and gender-matched controls, SATFAs were found to be significantly elevated in both male and female sub-groups. In the female sub-group analysis, significant increases were also observed for the arachidic acid and oleic acid. In the female AD sub-group, significantly lower levels were observed in α-linolenic acid, arachidonic acid and DHA. Overall, significant differences in fatty acids were predominantly observed in female MCI and AD cases. 

The mechanisms underlying the specific differences in the fatty acid profiles in relation to gender have not been clearly defined. However, it has been suggested that alterations in fatty acid profiles differ depending on the tissue being investigated and have been shown to be gender specific in both humans and rodents [94]. It has also been shown previously that such gender differences may be attributed to hormonal status, which directly affects the metabolism of fatty acids such as DHA [95]. Therefore, future studies involving a larger number of participants should focus on whether the risk for MCI and AD in males and females differ significantly with regard to the RBC fatty acids profile.

The study also showed that for the MCI group, the AUC was greater or equal to 0.80 for three of the six saturated fatty acids measured (behenic acid (C22:0; AUC = 0.80), stearic acid (C18:0; AUC = 0.80) and palmitic acid (C16:0; AUC = 0.85), two n-6 polyunsaturated fatty acids, namely γ-linolenic acid (C18:3n-6; AUC = 0.81) and arachidonic acid (20:4n-6; AUC = 0.84) and two n-3 PUFAs, namely α-linolenic acid (C18:3n-3; AUC = 0.81) and DHA (C22:6n-3; AUC = 0.83). This is suggestive that such a profile may prove to be useful as a series of potential diagnostic biomarkers reflecting increased risk for dementia. However, these potential biomarkers need to be further verified in additional larger prospective studies before their true potential diagnostic value can be determined. The biological significance of these profiles in relation to more advanced forms of dementia is also uncertain, such as AD, which was shown to have different RBC fatty acid profiles compared to MCI individuals. It would also be of interest to determine whether these profiles are specific to MCI and AD in relation to other forms of neurodegeneration, such as Parkinson’s disease or Huntingdon’s chorea. 

The current study further showed a positive correlation between MMSE score and the monounsaturated fat nervonic acid (C24:1n-9) in MCI individuals (r = 0.54, *p* = 0.01). The association of nervonic acid with MMSE may prove to be an interesting observation because nervonic acid is required in the synthesis of nerve cell myelin and was recently shown to be decreased in erythrocyte membranes in individuals at high risk of psychosis [96,97]. Furthermore, nervonic acid has been shown to be associated with preventive effects in obesity-related metabolic disorders [98]. It was also shown that the MMSE scores for the AD group had a negative correlation with the PUFA γ-linolenic acid (C18:3n-6) (r = −0.43, *p* = 0.05), which is in agreement with the observation that low levels of PUFAs are associated with cognitive impairment [99,100,101,102,103]. 

The RBC fatty acid profile presented in this study reflects the dietary patterns of elderly South Australians. Australian diets are primarily “western” in nature and consist of foods high in SATFAs and red meats but low in fresh fruits and vegetables, whole grains and seafood (accounting for the reduced PUFA values) [104,105]. Nutritional factors have previously been shown to have profound effects on cognitive performance. The result obtained from longitudinal studies has shown that diets high in SATFAs are correlated with cognitive decline, leading to an increased risk of developing MCI and AD [48,106,107,108,109]. It may be more relevant to follow the Mediterranean diet that contains larger proportions of both PUFAs and MUFAs, but reduced intake of SATFAs may result in reduced frequencies of cognitive decline, especially if adopted in those countries that traditionally follow the Western dietary pattern [43,110].

One of the limitations of the present study is the relatively small sample size. Although the effect size was large enough for most fatty acids to be measured at statistically significant levels, the study should be repeated in larger studies. A further limitation was that information on the measurement of depressive symptoms was not collected. This is important for the design of future studies as it has been shown that fatty acid profiles have been associated with varying degrees of depression, particularly in middle-aged and older adults [111,112]. An additional limitation was that an individual’s years of education were not available for the study subjects. The design of future studies should include these measurements as it has been shown that the level of education is a well-known risk factor for AD, which may, in turn, influence dietary habits [113,114]. It would also have been interesting to include more information on the health status of the participants, such as body mass index, clinical histories for hyperlipidemia and diabetes, as well as haemoglobin and haematocrit measurements, and especially their dietary pattern. One of the strengths of the current study is the fact that all MCI and AD cases were newly diagnosed and were recruited to take part in the study prior to any medical treatment. Participants were not on medication that may have proved to be a potential confounding factor, such as cholesterol-lowering drugs or the unknown effects that anticholinesterases may have on lipid profiles. 

More research is also needed to further test emerging evidence suggesting that the RBC fatty acid profile correlates with that of the brain and that changes in DHA concentration in neural cells may affect the processing of APP and the generation of amyloid-β in neurons [40,89,90,115]. Furthermore, recent studies indicate that the gut microbiome may play an important role in the metabolism and bioavailability of micronutrients, including fatty acids that are associated with the risk of AD. There is, therefore, a need to better understand the extent to which differences in RBC fatty acid profiles in MCI and AD cases are affected by variations in the microbiome [116,117,118,119]. 

As a future direction, it may be useful to (i) investigate whether the RBC fatty acid profiles can effectively and reliably identify those individuals who are at high risk of developing MCI to target them for primary prevention, including longitudinal studies, (ii) investigate the relationship between RBC fatty acid profiles and genotypes that may affect lipid metabolism, e.g., carriers of APOEε4 allele and polymorphisms in genes required for omega-3 and omega-6 fatty acid synthesis, such as the FADs 1 and FADs 2 genes [40,77,120] and (iii) test in intervention studies that are aimed at improving cognitive function in MCI’s which fatty acids in RBCs are best correlated with better outcomes. 

## 4. Materials and Methods

### 4.1. Study Design, Recruitment and Characteristics of Participants

This study was part of the South Australian Neurodegeneration, Nutrition and DNA Damage (SAND) study. Approval for the SAND study was obtained from CSIRO, Adelaide University and Calvary Hospital Ethics Committees. Twenty newly clinically diagnosed MCI (age 58–89 years) and 19 newly clinically diagnosed AD patients (age 46–89 years) not receiving anti-folate therapy or cancer treatment were recruited for the study. A power calculation based on standard deviations (±1.12) was performed based on the results of a previous RBC fatty acid experiment that measured levels of DHA and showed that a sample size of 20 individuals in each group has 90% power to detect a difference between DHA means of 0.84 with a significance level of (alpha) of 0.05 (two-tailed) [115,121]. MCI and AD patients were recruited at the Calvary Hospital, Walkerville, Adelaide, South Australia. All cases (MCI and AD) did not receive any treatment at the time of recruitment and providing blood samples. AD diagnosis was made by clinicians as per the criteria outlined by the National Institute of Neurological and Communicative Disorders and Stroke-Alzheimer’s Disease and related Disorders Association (NINCDS-AD&DA) [122]. Thirty-nine healthy controls with no medical issues were matched for age (age 46–90 years) and gender with subjects in the MCI and AD groups and were recruited using the CSIRO clinic database. Prior to consent, control subjects were screened for eligibility using the following criteria: healthy males/females, not taking any supplements with regard to the above recommended daily intake of micronutrients (required for genome maintenance, e.g., folate, vitamin B12), not clinically diagnosed with MCI or AD and have no family history of cognitive impairment (MCI and/or AD). Consented participants were required to complete the MMSE [111]. Participants in each study group did not differ significantly with regard to age and gender MMSE scores were found to be significantly lower in AD and MCI relative to controls and lower in AD relative to MCI. Age, gender and MMSE scores of the study groups are shown in Table 3. Additional patient information, including body mass index, clinical histories for hyperlipidemia and diabetes, as well as haemoglobin and haematocrit measurements were not available for further analysis.

### 4.2. Blood Collection

Prior to sample collection, a brief information session summarising the aims and procedures of the study was given to participants. Legal guardians/carers signed the consent form to provide tissue samples on behalf of the participants who were diagnosed with MCI and AD. A non-fasted blood sample was collected from consented newly clinically diagnosed MCI or AD patients prior to medical treatment and age and gender-matched controls by trained nursing staff at Calvary Hospital, Walkerville, Adelaide and the CSIRO clinic. A single blood sample (8 mL) was collected by venipuncture in an ethylenediamine tetra-acetic acid (EDTA) tube, placed on ice and then transported to the CSIRO laboratory. Whole blood tubes were centrifuged (Rotanda 460R, Adelab Scientific, Adelaide, Australia) at 1970× *g* for 20 min at 4 °C. Plasma was separated, and the total volume was maintained by adding 0.15 M Sodium Chloride (NaCl) solution to the remaining RBCs, mixed gently by inversion and centrifuged again at 1970× *g* for 10 min at 4 °C. The supernatant was discarded, and this step was repeated one more time. The supernatant was removed and replaced with the same amount of sterile milli-Q water (18.2 Ω resistivity, Millipore Milli-Q, Adelab Scientific, Australia), aliquoted (3 × 1.2 mL) and stored at −20 °C until analysis. Samples were analysed for saturated, monounsaturated, polyunsaturated and long-chain omega-3 polyunsaturated fatty acids, as described below.

### 4.3. Fatty Acid Analysis

RBC fatty acid analysis was performed under code on all frozen samples as per the modified method of Ridges et al. [123] and used the one-step extraction and transesterification protocol of Lepage et al. [124] and performed as described previously [125,126]. Briefly, 2 mL of methanol (Merck, Sydney, Australia) and toluene (Merck, Sydney, Australia (4:1 *v*/*v*) were mixed into each thawed sample. Next, 200 µL of acetyl chloride (Fluka, Sydney, Australia) was added dropwise whilst vortexing the sample. Samples were then incubated at 100 °C for an hour. Tubes were allowed to cool at room temperature for 5 min and then immersed in cold water for 5 min. Then, 3 mL of 10% potassium carbonate (Merck, Sydney, Australia) was slowly added to the contents, followed by 200 µL of toluene. The samples were vigorously shaken for 60 s before centrifugation at room temperature for 5 min at 1640× *g*. The upper toluene phase containing fatty acid methyl esters was transferred to another disposable 12 mm × 75 mm glass tube and evaporated till dry under a stream of nitrogen in a dry block heater (35–40 °C). Hexane (1 mL; Merck, Sydney, Australia) was then added to the tube contents to dissolve the esters. The tube contents were transferred into columns that were pre-washed with 1.0 mL hexane, and the methyl esters eluted with 2 × 1 mL 10% diethyl ether in hexane and collected into conical chromatography vials (Agilent Technologies, Santa Clara, CA, USA). The fatty acid methyl esters were analysed by gas chromatography using a 30 mm × 0.53 internal diameter capillary column (Agilent Technologies, Santa Clara, CA, USA). 

ChemStation software LTS 01.11 was used to measure peak areas (Agilent Technologies, Australia). The results for each fatty acid were expressed as a percentage (%) of the total area of the known fatty acid peaks. The identification of these peaks was ascertained based on a comparison of retention times using Supelco 37 component FAME mix (47885-U) or 68A (with added EPA) purchased from NuchekPrep Inc (Elysian, MN, USA 56028-0295).

### 4.4. Statistical Analysis

All results for each fatty acid are expressed as mean ± standard deviation (SD) and were calculated as a percentage of total RBC cell fatty acids. One-way analysis of variance (ANOVA) was used to determine the significance of the parameters being measured across all three groups, following a Bonferroni correction for multiple pairwise comparisons made using a two-tailed paired t-test for Gaussian distribution and the Wilcoxon test for non-Gaussian distribution. Receiver operating characteristic (ROC) curves and their AUC were determined for both MCI and AD. ROC curve analysis was performed as a tool to evaluate whether RBC fatty acids may prove to be useful as a potential diagnostic test. The AUC is a measure of how useful a parameter may be in distinguishing between our control, MCI and AD groups. Sensitivity, specificity and likelihood ratios were determined using arbitrarily chosen cut-off values to maximise both specificity and sensitivity values. Correlation analyses between MMSE scores and RBC fatty acid components were also determined for three study groups. All statistical analyses were performed using GraphPad Prism 6 (GraphPad Inc., San Diego, CA, USA) and SPSS (version 20) for one-way ANOVA and multiple regression models, including general linear modelling. Differences between groups were considered statistically significant at *p* < 0.05.

## 5. Conclusions

The results of our study support (i) the hypothesis that analysis of RBC fatty acids can be useful in identifying individuals at high risk of developing MCI and AD and (ii) that more research is needed to verify these results.

## Figures and Tables

**Figure 1 ijms-24-14164-f001:**
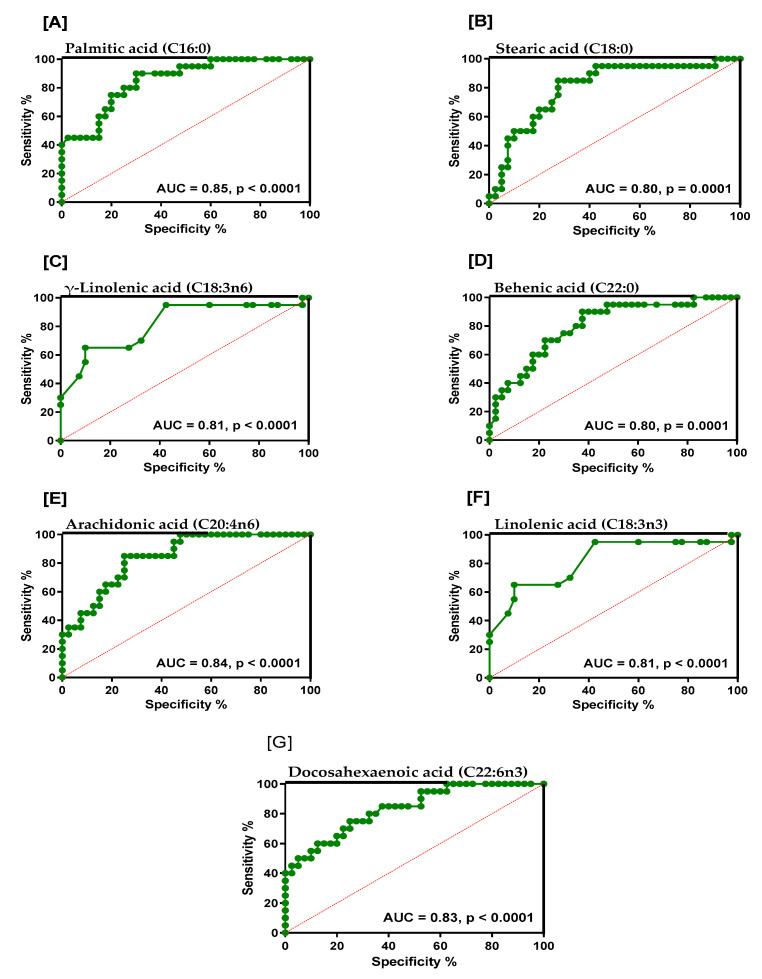
Graphs showing area under the curve (AUC > 0.8) and *p*-values for (**A**) palmitic acid, (**B**) stearic acid, (**C**) γ-linolenic acid, (**D**) behenic acid, (**E**) arachidonic acid, (**F**) linolenic acid and (**G**) docosahexaenoic acid in the MCI cohort.

**Table 1 ijms-24-14164-t001:** Red blood cell fatty acid analysis (wt% of total) from mild cognitive impairment, Alzheimer’s disease and normal age and gender-matched controls.

Fatty Acids	Controls	MCI	AD	*p* Value (ANOVA)
**SATFAs**				
C14:0 (Myristic acid)	0.32 ± 0.15	0.41 ± 0.10	0.36 ± 1.4	*0.061*
C15:0 (Pentadecanoic acid)	0.34 ± 0.13	0.36 ± 0.10	0.31 ± 0.1	*0.67*
C16:0 (Palmitic acid)	21.07 ± 2.88 ^a^	25.48 ± 3.40 ^b^	23.36 ± 3.86 ^b^	** *<0.00001* **
C18:0 (Stearic acid)	18.00 ± 2.17 ^a^	20.61 ± 2.32 ^b^	18.73 ± 2.11 ^a^	** *0.0001* **
C20:0 (Arachidic acid)	0.71 ± 0.1 ^a^	0.81 ± 0.11 ^b^	0.74 ± 0.09 ^a^	** *0.003* **
C22:0 (Behenic acid)	2.60 ± 0.41 ^a^	3.13 ± 0.51 ^b^	2.66 ± 0.53 ^a^	** *0.0002* **
C23:0 (Tricosanoic acid)	0.39 ± 0.08 ^a^	0.47 ± 0.07 ^b^	0.38 ± 0.10 ^a^	** *0.007* **
C24:0 (Lignoceric acid)	8.70 ± 0.99 ^a^	9.76 ± 1.16 ^b^	8.69 ± 1.04 ^a^	** *0.001* **
Total SATFAs	52.14 ± 6.17 ^a^	61.05 ± 7.14 ^b^	55.26 ± 7.34 ^a^	** *0.00002* **
**MUFAs**				
C16:1n-9 (Palmitoleic acid)	0.30 ± 0.10	0.31 ± 0.09	0.34 ± 0.30	*0.64*
C18:1n-9 (Oleic acid)	14.88 ± 1.10 ^a^	16.08 ± 1.15 ^b^	16.06 ± 1.63 ^b^	** *0.002* **
C22:1n-9 (Erucic acid)	0.15 ± 0.03	0.13 ± 0.02	0.14 ± 0.03	*0.08*
C24:1n-9 (Nervonic acid)	6.86 ± 0.72	7.18 ± 0.84	6.8 ± 0.93	*0.29*
Total MUFAs	22.20 ± 1.54 ^a^	23.72 ± 1.58 ^b^	23.39 ± 2.57 ^ab^	** *0.012* **
**n-6 PUFAs**				
C18:2n-6 (Linoleic acid)	6.11 ± 1.22 ^a^	4.76 ± 1.38 ^b^	5.40 ± 1.36 ^ab^	** *0.001* **
C18:3n-6 (γ-Linolenic acid)	0.09 ± 0.02 ^a^	0.06 ± 0.02 ^b^	0.08 ± 0.05 ^ab^	** *0.03* **
C20:3n-6 (Eicosatrienoic acid)	1.05 ± 0.34 ^a^	0.74 ± 0.42 ^b^	0.92 ± 0.34 ^ab^	** *0.009* **
C20:4n-6 (Arachidonic acid)	10.18 ± 3.39 ^a^	5.54 ± 3.34 ^b^	8.41 ± 4.19 ^a^	** *0.00004* **
Total n-6 PUFA	17.44 ± 4.40 ^a^	11.12 ± 5.1 ^b^	14.83 ± 5.40 ^a^	** *0.00003* **
**n-3 PUFAs**				
C18:3n-3 (α-Linolenic acid)	0.08 ± 0.03 ^a^	0.03 ± 0.03 ^b^	0.05 ± 0.04 ^b^	** *0.0005* **
C20:5n-3 (Eicosapentaenoic acid)	1.01 ± 0.69	0.49 ± 0.50	0.96 ± 1.19	*0.06*
C22:6n-3 (Docosahexaenoic acid)	5.07 ± 2.06 ^a^	2.47 ± 1.77 ^b^	3.75 ± 2.13 ^b^	** *0.00003* **
Total n-3 PUFAs	6.17 ± 2.60 ^a^	3.00 ± 2.25 ^b^	4.77 ± 3.17 ^ab^	** *0.0002* **
Total PUFAs	23.61 ± 6.55 ^a^	14.13 ± 7.16 ^b^	19.60 ± 8.01 ^a^	** *<0.00002* **
n-3/n-6 ratio	0.35	0.27	0.32	** *0.004* **
Total PUFA/Total SATFA ratio	0.45	0.23	0.35	** *0.00001* **

Values are reported as mean ± SD for N = 39 (controls), N = 20 (MCI), and N = 19 (AD). Groups sharing the same letters are not significantly different. Abbreviations: SATFA, saturated fatty acid; MUFA, monounsaturated fatty acid; PUFA, polyunsaturated fatty acid; Total n-3, Total Omega-3 fatty acids; Total n-6, Total Omega-6 fatty acids. Values in same row that have different letters are statistically significantly different (*p* < 0.05) from each other.

**Table 2 ijms-24-14164-t002:** Pairwise comparison for MCI and age and gender-matched controls and AD and age and gender-matched controls.

Fatty Acids	MCI Mean ± SEM	Con MCI Mean ± SEM	*p* Value	AD Mean ± SEM	Con AD Mean ± SEM	*p* Value
**14:0** (Myristic acid)	0.42 ± 0.02	0.37 ± 0.04	*0.41*	0.36 ± 0.03	0.28 ± 0.01	** *0.03* **
**15:0** (Pentadecanoic acid)	0.36 ± 0.03	0.35 ± 0.03	*0.92*	0.32 ± 0.03	0.32 ± 0.03	*0.99*
**16:0** (Palmitic acid)	25.48 ± 0.76	22.20 ± 0.77	** *0.003* **	23.36 ± 0.87	19.94 ± 0.35	** *0.0004* **
**18:0** (Stearic acid)	20.61 ± 0.52	19.09 ± 0.56	*0.06*	18.73 ± 0.47	16.91 ± 0.20	** *0.001* **
**20:0** (Arachidic acid)	0.81 ± 0.03	0.75 ± 0.03	*0.11*	0.74 ± 0.02	0.68 ± 0.02	** *0.01* **
**22:0** (Behenic acid)	3.14 ± 0.12	2.78 ± 0.11	*0.06*	2.66 ± 0.12	2.43 ± 0.06	*0.08*
**23:0** (Tricosanoic acid)	0.47 ± 0.02	0.41 ± 0.02	*0.11*	0.38 ± 0.01	0.38 ± 0.01	*0.69*
**24:0** (Lignoceric acid)	9.77 ± 0.26	8.99 ± 0.27	*0.08*	8.70 ± 0.23	8.42 ± 0.14	*0.34*
**Total SATFA**	61.05 ± 1.60	54.94 ± 1.67	** *0.02* **	55.26 ± 1.64	49.35 ± 0.54	** *0.001* **
**16:1n-9** (Palmitoleic acid)	0.32 ± 0.02	0.31 ± 0.02	*0.87*	0.35 ± 0.07	0.29 ± 0.02	*0.47*
**18:1n-9** (Oleic acid)	16.08 ± 0.24	15.37 ± 0.27	*0.07*	16.06 ± 0.37	14.38 ± 0.16	** *0.001* **
**22:1n-9** (Erucic acid)	0.14 ± 0.01	0.15 ± 0.01	*0.07*	0.15 ± 0.01	0.17 ± 0.01	*0.15*
**24:1n-9** (Nervonic acid)	7.19 ± 0.20	7.06 ± 0.12	*0.56*	6.83 ± 0.21	6.67 ± 0.19	*0.54*
**Total MUFA**	23.72 ± 0.35	22.89 ± 0.33	*0.11*	23.39 ± 0.58	21.50 ± 0.29	** *0.01* **
**18:2n-6** (Linoleic acid)	4.76 ± 0.31	5.74 ± 0.30	*0.06*	5.41 ± 0.31	6.49 ± 0.22	** *0.007* **
**18:3n-6** (γ-linolenic acid)	0.07 ± 0.01	0.09 ± 0.01	** *0.02* **	0.08 ± 0.01	0.10 ± 0.00	*0.28*
**18:3n-3** (α-linolenic acid)	0.04 ± 0.01	0.07 ± 0.01	** *0.005* **	0.05 ± 0.01	0.09 ± 0.01	** *0.007* **
**20:3n-6** (Eicosatrienoic acid)	0.75 ± 0.09	0.94 ± 0.08	*0.12*	0.93 ± 0.08	1.17 ± 0.07	** *0.02* **
**20:4n-6** (Arachidonic acid)	5.55 ± 0.77	8.61 ± 0.89	** *0.01* **	8.41 ± 0.94	11.75 ± 0.35	** *0.001* **
**20:5n-3** (Eicosapentaenoic acid)	0.49 ± 0.11	0.84 ± 0.12	*0.06*	0.96 ± 0.27	1.19 ± 0.18	*0.52*
**22:6n-3** Docosahexaenoic acid)	2.48 ± 0.40	4.20 ± 0.49	** *0.02* **	3.76 ± 0.48	5.96 ± 0.35	** *0.001* **
**Total PUFA**	15.17 ± 1.7	22.05 ± 1.88	** *0.02* **	21.30 ± 2.02	29.02 ± 0.74	** *0.001* **
**Total n-3 PUFA**	4.05 ± 0.70	6.67 ± 0.74	** *0.02* **	6.47 ± 0.94	9.51 ± 0.58	** *0.009* **
**Total n-6 PUFA**	11.12 ± 1.15	15.38 ± 1.18	** *0.02* **	14.83 ± 1.22	19.51 ± 0.49	** *0.001* **
**n-3/n-6 ratio**	0.32 ± 0.03	0.41 ± 0.02	** *0.02* **	0.40 ± 0.04	0.49 ± 0.03	*0.11*
**PUFA/SATFA**	0.27 ± 0.04	0.43 ± 0.06	** *0.01* **	0.41 ± 0.05	0.59 ± 0.02	** *0.001* **

**Table 3 ijms-24-14164-t003:** Age, gender ratio and MMSE scores in study groups.

	Controls N = 39	MCI N = 20	AD N = 19
**Age (years)**			
Mean ± S.D	75.75 ± 9.95	74.60 ± 8.74	76.85 ± 10.96
Range	46–90	58–89	46–89
**Gender**			
Male	12	7	4
Female	28	13	15
**MMSE (0–30)**			
Mean ± S.D	28.56 ± 1.33 ^a^	26.25 ± 2.33 ^b^	21.0 ± 3.86 ^c^
Range	25–30	21–29	14–26

Significant at *p* < 0.05. Groups not showing the same letter are significantly different from each other.

## Data Availability

Data will be uploaded to a publicly available repository upon acceptance of the manuscript.

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
