# Peer review of "Red Blood Cell Fatty Acid Profiles Are Significantly Altered in South Australian Mild Cognitive Impairment and Alzheimer’s Disease Cases Compared to Matched Controls"

_ijms, 2023, doi:10.3390/ijms241814164_

Round 1

Reviewer 1 Report

The manuscript can be accepted in present form.

Author Response

Attached please find our point-to-point response to the Reviwewer's comment 

Reviewer 2 Report

Research work on Red blood cell fatty acid profiles are significantly altered in South Australian mild cognitive impairment and Alzheimer’s disease cases compared to matched controls recommended for acceptance as communication  after minor revision. 

Comment 1:  Authors have mentioned it has also been suggested that omega-3 may alter membrane stability as well as amyloid precursor protein (APP) cleavage by β-secretase, thereby reducing the production of beta-amyloid (β-amyloid) [83]. It will be better if authors show the pathway for this and mention recent references.

Comment 2: Include recent references in the introduction part.

Moderate editing of English language required

Author Response

Attached please find our point-to point response to the Reviewer's comment 

Reviewer 3 Report

The authors present the article entitled "Red blood cell fatty acid profiles are significantly altered in South Australian mild cognitive impairment and Alzheimer's disease cases compared to matched controls." in which they try to establish a correlation between the profile of fatty acids in red cells and the risk of developing AD and MCI.

In general, the article tries to resolve the hypothesis with the results they obtained, however, my main concern is the size of their sample, as they mention at the end of their discussion. In this sense, I consider that the relevance of the study is as great as it would have been with a larger sample, in addition, the relevance of its results is evident in the title of the article, it is probably not as significant as mentioned.
The other great limitation, as mentioned by the authors, was the lack of knowledge of the educational level of the participants, which is directly related to MCI and AD, this could generate some bias in the study.

It would also have been interesting to include more information on the health status of the participants, such as body mass index, clinical histories for hyperlipidemia and diabetes as well as haemoglobin and haematocrit measurements, and especially their diet. Another important factor would be for the authors to try to discuss whether population variations could influence the fatty acid profile of red blood cells, beyond differences in diet.

Finally, it would be interesting for the authors to discuss how the lipid profile reported in red blood cells would be related to the plasticity and functions of the central nervous system cells.

Moderate editing of English language required

Author Response

(The authors gave the same response as above.)

Reviewer 4 Report

In the Methods section, line 354-356 MMSE icluding... can be deleted, since MMSE score is clearly described in numerous earlier studies.

In the Discussion / Conclusion section, the ApoE sttus should be basically discussed, citing papers as Martinsen et al, Nutrients 2023;15(9): 2023 and others...

Furtheron, Dietary fats, gut microbiota and dementia syndromes should be discissed as well, citing papers like Leblhuber et al, Nutrients 2021, 13(2):361 and others...

Author Response

(The authors gave the same response as above.)

Round 2

Reviewer 3 Report

The article can be accepted in the present form.